# StressME: Unified computing framework of *Escherichia coli* metabolism, gene expression, and stress responses

**Jiao Zhao**[1], **Ke Chen**[2], **Bernhard O. Palsson**[2], **Laurence Yang**[1]*

**1** Department of Chemical Engineering, Queen's University, Kingston, Ontario, Canada, **2** Department of Bioengineering, University of California, San Diego, La Jolla, California, United States of America

* laurence.yang@queensu.ca

**Data Availability Statement:** The authors confirm that all data underlying the findings are fully available without restriction. All relevant data are within the paper, its Supporting Information files,

## Abstract

Generalist microbes have adapted to a multitude of environmental stresses through their integrated stress response system. Individual stress responses have been quantified by *E. coli* metabolism and expression (ME) models under thermal, oxidative and acid stress, respectively. However, the systematic quantification of cross-stress & cross-talk among these stress responses remains lacking. Here, we present StressME: the unified stress response model of *E. coli* combining thermal (FoldME), oxidative (OxidizeME) and acid (AcidifyME) stress responses. StressME is the most up to date ME model for *E. coli* and it reproduces all published single-stress ME models. Additionally, it includes refined rate constants to improve prediction accuracy for wild-type and stress-evolved strains. StressME revealed certain optimal proteome allocation strategies associated with cross-stress and cross-talk responses. These stress-optimal proteomes were shaped by trade-offs between protective vs. metabolic enzymes; cytoplasmic vs. periplasmic chaperones; and expression of stress-specific proteins. As StressME is tuned to compute metabolic and gene expression responses under mild acid, oxidative, and thermal stresses, it is useful for engineering and health applications. The modular design of our open-source package also facilitates model expansion (e.g., to new stress mechanisms) by the computational biology community.

## Author summary

A fundamental understanding of multi-stress adaptation in *E.coli* has potential industrial relevance. While individual stress responses have been quantified through the protein regulatory network in *E.coli*, the systematic quantification of the cross-stress & cross-talk among stress responses remains lacking. Here, we develop a new modeling pipeline by which thermal, oxidative and acid stress response can be coupled to each other, and the metabolic activities, protein and metabolic flux redistribution due to cross-stress & cross-talk can be quantified. We optimize the effective rate constants in the integrated model. We then confirm the model robustness by validating against the published data under single stress. Finally, we use the model to characterize the cross-adaptation between protective and catalytic proteins as well as between chaperones present in different cellular

Github (https://github.com/QCSB/StressME) and Docker Hub (https://hub.docker.com/r/queensysbio/stressme/tags).

**Funding:** This work was supported by the National Institute of General Medical Sciences of the National Institutes of Health Grant R01GM057089 to BOP and LY, the Natural Sciences and Engineering Research Council of Canada (NSERC) [RGPIN-2020-06325], the Government of Canada through Genome Canada and Ontario Genomics (OGI-207), the Government of Ontario through an Ontario Research Fund (ORF), and Queen's University to LY. This research was enabled in part by support from Centre for Advanced Computing (Queen's University), Compute Ontario (computeontario.ca) and the Digital Research Alliance of Canada (alliancecan.ca) to LY. The funders had no role in study design, data collection and analysis, decision to publish, or preparation of the manuscript.

**Competing interests:** The authors have declared that no competing interests exist.

compartments. We find effective cross-protection against cross stress by adapting the *E. coli* cells to the thermal stress first. We also indicate the presence of cross-talk through trade-offs by which the cell may refuse to give up more protein allocation away from one stress response to the other, because doing so would decrease stress tolerance further. The single stress plug-in design makes the model build-up pipeline flexible and expandable, allowing incorporation of more stressors into the model architecture for industrial applications.

## Introduction

Microbes in fluctuating environments have adapted to environmental changes by combining individual stress detection and response systems into an integrated network. For a better understanding of how microorganisms manage to resist environmental stresses, much work has been focused on intricate mechanisms of a single stress response at the physiological and molecular levels.

Among these studies, *E.coli* is an excellent model bacteria because the whole genome sequencing has been completed and the genome structure and function for the single stress response has been characterized. It has been found that *E.coli* cells can resist osmotic stress by overexpressing methionine-related genes *metK* and *mmuP*, and inactivating the stress-related gene *bolA* [1]. To survive the starvation stress, levels of RpoS regulon gene expression in *E.coli* are increased upon glucose starvation, but RpoS levels are only slightly increased for ammonia starvation and much lower than those detected in glucose starvation [2]. It has been confirmed that the stability of RpoS proteins affected by the level of proteolysis under carbon starvation is responsible for RpoS regulation [3]. *E.coli* can enhance acid tolerance by periplasmic HdeA/HdeB chaperones to prevent periplasmic proteins from aggregation under acidic conditions [4]. Another mechanism of increasing acid resistance in *E.coli* is to increase production of unsaturated fatty acids of the membrane, thus decreasing the membrane fluidity for acid tolerance [5]. To tolerate thermal stress, *E.coli* cells use both DnaK chaperone machine and GroEL/S chaperonin to help other proteins to properly fold and not aggregate at higher temperatures [6]. Two stimulons (peroxide stimulon and superoxide stimulon), each containing more than 30 genes, have been characterized in *E.coli* under oxidative stress [7]. Some of these genes constitute the OxyR and SoxRS regulons, with gene products responsible for either prevention (catalases and superoxide dismutases) or repair (endonuclease) of oxidative damage.

Despite a good understanding of the physiological and molecular responses to individual environmental stress, little work has been done to establish an integrated network system that connects the molecular targets of environmental stressors to other intracellular molecules leading to collective physiological responses. To address these limitations, three single stress network models based on genome-scale models of metabolism and macromolecular expression (ME-models) for *E.coli* [8,9] have been developed to describe the fundamental mechanisms by which cells respond to single thermal [10], acid [11] and oxidative [12] stress independently.

The thermal-stress-response model, called FoldME [10], was used to describe the reallocation of the proteomics resources over the intracellular network at higher temperatures. Such a response at the molecular level can stabilize proteins by boosting the concentration of protective proteins (chaperones) in cytosol. The AcidifyME [11], on the other hand, established a quantitative framework with focus on the acid-stress response from membrane and periplasmic proteins. These target proteins, which are more exposed to external acid stress than cytoplasmic proteins, are associated with lipid fatty acid composition and enzyme activities in

membrane, and the periplasmic chaperone protection mechanisms. The OxidizeME [12] was reconstructed to account for the damage to metalloproteins in cytosol by reactive oxygen species (ROS), including the fundamental mechanisms of demetallation/mismetallation of Fe(II) proteins with alternative metal ions, damage and repair of iron–sulfur clusters found in metalloproteins, and damage to DNA by hydroxyl radicals from spontaneous reaction of Fe(II) with $H_2O_2$.

What is lacking is a good understanding of the interactions that occur between different stress response systems in *E. coli* to react collectively to multiple environmental stresses. While single stress response can be quantified by ME models through the overall protein regulatory network [10–12], and some phenotypes in response to cross-stress can be characterized in the wet lab [13], the actual quantification of the response to multiple stressors in *E.coli* remains lacking, but such information would be very helpful for industrial microbial processes. For example, during food industrial processes, foodborne bacteria may be exposed to a variety of environmental stresses at one time or by order, such as acid stress from preservatives [14], oxidative stress from food sterilization (e.g., cold plasma) [15], and thermal stress from mild heat treatment [16]. Additionally, during microbial waste treatment and high-density cultivation for microbial products, cells may have a higher chance of experiencing a set of environmental stressors simultaneously, such as thermal, acid, and oxidative stress caused by both endogenous and exogenous factors [17,18]. A better understanding of the cross-adaptation mechanisms by modeling approach will be helpful for tailoring the optimal control strategy to specific biotechnological processes.

To address these limitations, we have developed a StressME modeling framework by which single-stress systems can be coupled to each other, and phenotypes, proteome and fluxome in response to multiple stressors can be quantified. Mild acidic pH, heat treatment and ROS exposure relevant to industrial biotechnological processes were chosen in StressME simulations to study the cross-adaptation among different stresses. Hence, the StressME model and simulations in this study would have potential industrial relevance. They can be used as a benchmark for further prediction studies involving more environmental stressors experienced in industrial settings.

## Results and discussion

### StressME model overview

Starting from the *E. coli* K-12 MG1655 ME-model (EcoliME) iJL1678-ME [9], this work used the Ecolime framework to build an integrated StressME model, called iZY1689-StressME, from three published single-stress ME models [10–12].

We built StressME using the latest ME codebase, COBRAme [9]—noted here since FoldME [10] (thermal stress response) was published with a previous version of the ME modeling codebase. Starting from COBRAme, and its associated ME model of *E. coli* K-12 MG1655 (iJL1678b-ME), we integrated FoldME [10], AcidifyME [11] and OxidizeME [12]. The resulting model, called iZY1689-StressME, comprises 1,689 genes, 1,578 proteins, 1,673 metabolites, 1,692 complexes and 36,735 reactions (Fig 1). This biological scope is a significant expansion over the original iJL1678b-ME composed of 1,678 genes, 1,568 proteins, 1,671 metabolites, 1,526 complexes and 12,655 reactions. More details about the structure difference between StressME and iJL1678b-ME, and the reactions added to StressME after every reconstruction step are shown in Tables A and B in S1 Appendix. Chemical equations for each added reaction can be found in S2 Appendix.

In single-stress ME models [10–12], the effect of a stress on the cell growth was quantified by a change in the properties of the affected cellular proteins and a reallocation of the

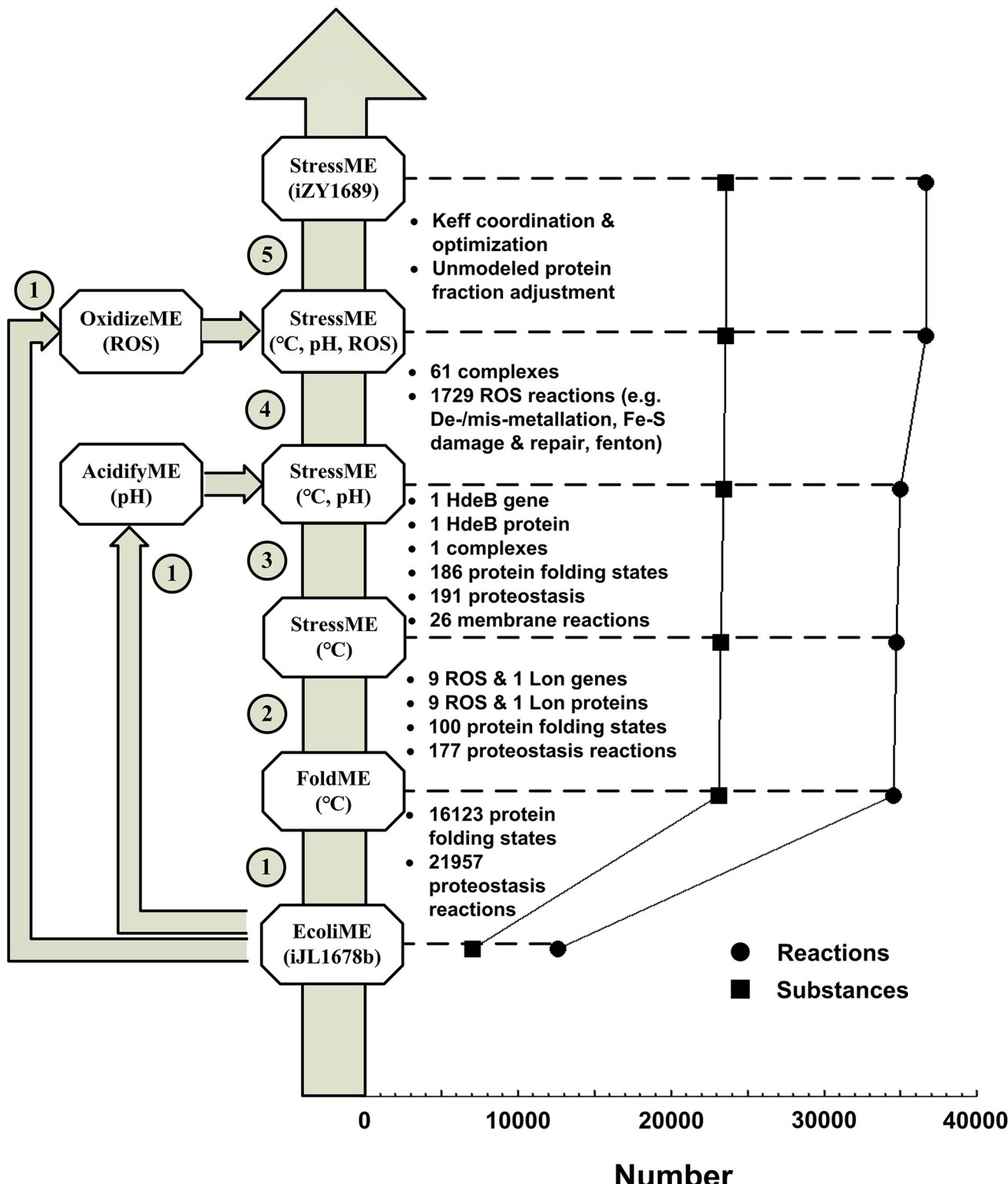

**Fig 1. Pipeline and statistics for building the StressME (iZY1689) model.** ① EcoliME (iJL1678b) to FoldME, OxidizeME and AcidifyME in python 3.6 ② FoldME to single StressME (˚C) ③ Single StressME integrated with AcidifyME to dual StressME (˚C and pH) ④ Dual StressME integrated with OxidizeME to Triple StressME (˚C, pH and ROS) ⑤ Triple StressME optimization to final iZY1689.

proteome in response to that property alteration. Thus, all proteins described by one single-stress ME model but ignored by the other should be included in the integrated StressME. Totally eleven single-stress-associated proteins were therefore coupled to the EcoliME framework based on their subcellular locations. Among them, nine cytoplasmic proteins in response to oxidative stress and one (ATP-dependent protease, Lon) associated with misfolded protein degradation were added to the published FoldME [10] (S3 Appendix).

The single-stress plug-in design (see Materials and Methods for detail) makes the StressME build-up pipeline flexible and expandable, allowing the model to incorporate more stress modules (e.g., starvation, osmotic, alkaline, and cold stresses) to the EcoliME-model architecture in the future.

Average time was estimated for the StressME running in the local computer (Intel Core i7-10875H @ 2.30 GHz, 8 cores, 32GB DDR4-2933MHz) and Graham heterogeneous cluster (Compute Canada, 2 x Intel E5-2683 v4 Broadwell @ 2.1GHz, 1 CPU; 8 GB memory per task). The typical computation time for running a nonlinear programming solver (quad-precision MINOS) for the relevant use cases of the StressME model is listed in Tables A and B in S4 Appendix. It shows that by using the warm-start option in MINOS, the average time for running StressME in different platforms can be between 8 and 47 minutes depending on stress conditions. When growth rate can be fixed while maximizing metabolic, transcription, translation, or complex formation rates, the average time can be as fast as between 0.8 and 6 minutes.

## Quality control of the StressME model

The StressME build-up pipeline is based on independent single-stress ME models, with model parameters fit to experimental data obtained under specific strain (i.e., wild-type for OxidizeME and AcidifyME, and heat-evolved for FoldME) and stress (i.e., oxidative, acid and thermal) conditions. One example of the variability in parameter values across single-stress ME models is Keff, which is defined as the effective turnover rates for intracellular processes catalyzed by the corresponding macromolecules. Eq 1 (Methods section) indicates that, if a Keff value is increased, the amount of the protein to be synthesized is decreased. Hence, the difference in Keff between single-stress ME models may cause proteome reallocation in an integrated StressME, which complicates investigation of the response to multiple stresses by inherent mechanisms. For example, some essential macromolecules may require more proteomic resources to synthesize chaperones to assist them to reach their native fold. The uncoordinated Keff may cause overexpression of chaperones to protect these macromolecules from unfolding, which may affect other metabolic functions because of the competition for the shared proteome resources. Therefore, the first step to quality control of the StressME model was to calibrate the key uncoordinated Keffs so that their optimal values can correctly describe the proteome reallocation in response to different strain and stress conditions.

## Keff in single- and Integrated-StressME models

Over the past 10 years of development, ME-models have been published with differences in their Keff values [8,10–12,19]. With StressME, we combine three stress-ME models and include updates introduced in COBRAme [9]. Therefore, we sought to create a consolidated Keff vector enabling individual stress responses to be reproduced.

First, we identified incompatibilities between the Keff values used in FoldME and AcidifyME: swapping their Keff values produced simulations inconsistent with published results for thermal and acid responses (Fig 2C–2D). These inconsistencies are caused by differing Keff values used in FoldME and AcidifyME/OxidizeME: Keffs are more narrowly distributed in FoldME than AcidifyME and OxidizeME, and they differ in median value (Fig 2A and 2B).

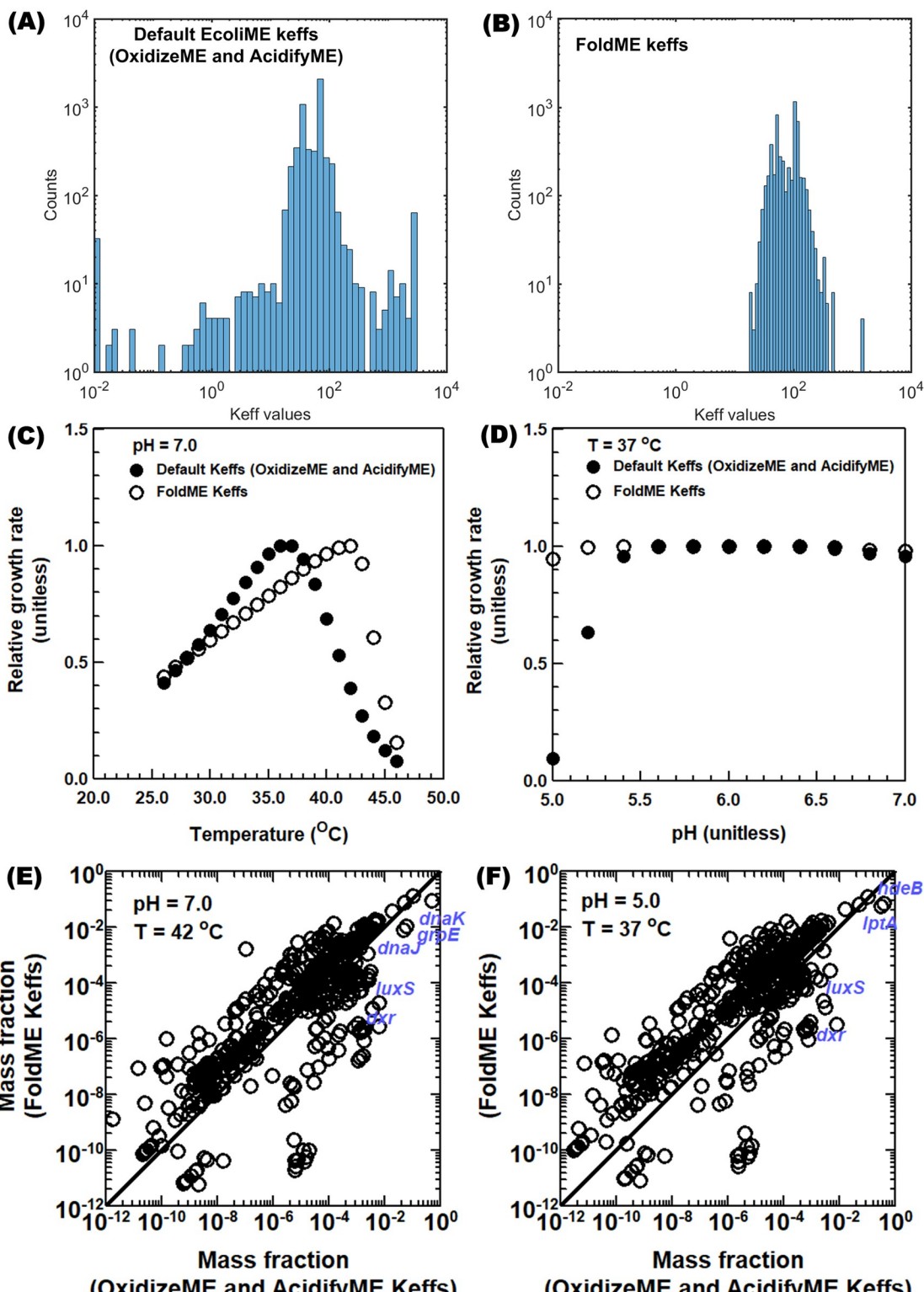

**Fig 2. Investigating sensitivity of predicted phenotypes and proteome to the kinetome (keff, effective rate constants).** (A) and (B): Distribution of Keffs for metabolic reactions used in previous single-stress ME models; (C) and (D): Effect of Keffs on growth rates simulated by StressME under different stress conditions; (E) and (F) Effect of Keffs on simulated protein mass fractions by StressME under different stress conditions.

Interestingly, simulated growth using FoldME Keffs produces more heat-evolved phenotypes, having max growth rate near 42˚C (Fig 2C). Meanwhile, using AcidifyME/OxidizeME Keffs in StressME produced temperature-dependent growth rates closely resembling wild-type *E. coli*–with max growth rate near 37˚C. Therefore, with respect to thermal response, StressME computes two phenotypes: heat-evolved (using FoldME Keffs) and wild-type (using AcidifyME/OxidizeME Keffs).

The simulated proteomes under thermal or acid stress also differ between the two Keff sets (Fig 2E and 2F). It shows that the FoldME Keff model has the capacity to express more other useful proteins (besides chaperones) because there is more room to reallocate the proteome toward proteins that increase growth rates under stress conditions (Fig 2C and 2D). A key reason for the increased protein allocation capacity is that the *dxr* protein can be expressed at a much lower (around 8872-fold) concentration (Eq 1) using the FoldME Keffs (88.72) vs. the wild type Keffs (0.01). Therefore, other proteins can be expressed at higher concentrations when *dxr* and its protective chaperones (*dnaK* and *dnaJ*) are expressed at lower concentrations in the FoldME Keff model, given the constraint on total proteome mass.

## Key rate constants differentiating stress response

We then sought to identify the key Keffs differentiating wild-type and heat-evolved stress responses. We used a simple, single effect sensitivity analysis, alternating between wild-type and heat-evolved Keff values sequentially for every reaction and computing the max growth rate. Growth was most sensitive to the Keff of DXPRli, catalyzed by 1-deoxy-D-xylulose 5-phosphate reductoisomerase (*dxr*) requiring either Mn2+, Co2+, or Mg2+ (Fig 3A). Changing the Keff for DXPRli between 0.01 to 88.72 resulted in growth rate increasing from 0.88 to 1.04 h$^{-1}$. Other reactions showed little to no growth rate change when the Keff was swapped between wild-type and heat-evolved values. These keff values may already be in a reasonable range, or some reactions may have alternate pathways to fulfill similar functions.

DXPRli is a key reaction in terpenoid biosynthesis. Terpenoid is essential for *E. coli* growth and metabolism, which is associated with electron transport (ubiquinone and menaquinone) in respiration chain and membrane biosynthesis and stability [20]. Fig 3B indicates that lower default Keff for DXPRli would significantly increase the synthesis of *dxr* and the protective chaperones (*dnaK* and *dnaJ*) for folding, as compared to the higher FoldME Keff. This would reduce the synthesis of other essential macromolecules under proteome constraint, resulting in low levels of metabolic activities.

We investigated whether heat-sensitive protein unfolding can explain DXPRli's heat sensitivity. To do so, we perturbed its associated protein unfolding Keq. Decreasing the folding requirement of *dxr* at higher temperature should increase the growth rate. This is confirmed by artificially lowering the unfolding Keq (Native ⇌ Unfolded) for *dxr*, which increased growth rates under thermal stress (Fig 3C). However, such an effect would be less significant at lower temperatures.

The second most sensitive keff is RHCCE. Lowering its keff greatly increases protein mass fraction (Fig 3D) but growth rate is not impacted much (Fig 3A), suggesting that there are alternate pathways to fulfill some function of RHCCE, so that not much proteomic resource is required to synthesize the corresponding protective chaperones.

In StressME (reaction and metabolite identifiers are standardized by BIGG identifiers [16]), the RHCCE reaction is catalyzed by *S*-ribosylhomocysteine cleavage enzyme (encoded by gene *luxS*) to yield homocysteine for methionine synthesis.

RHCCE (S-ribosylhomocysteine cleavage enzyme)

$$rhcys\_c \rightarrow dhptd\_c + hcys\_L\_c$$

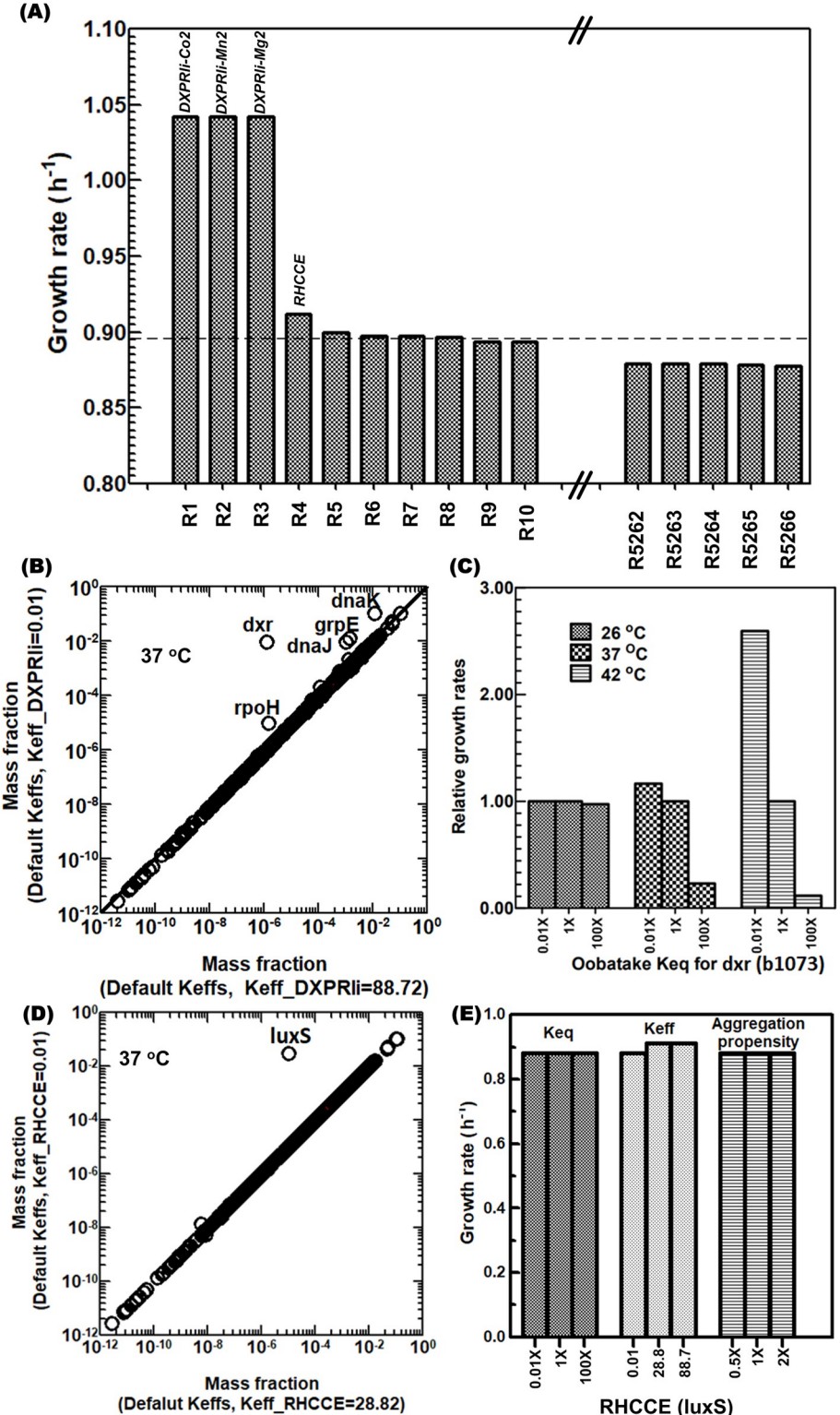

**Fig 3. Stress-evolved Keffs for StressME** (A) Sensitivity of simulated growth rates to keff choice for each reaction. Two reactions are affected the most: reaction DXPRli (1-deoxy-D-xylulose 5-phosphate reductoisomerase, dxr); reaction RHCCE (S-ribosylhomocysteine lyase, luxS) (B) Effect of keff for the reaction DXPRli on proteome allocation (C) Effect of protein unfolding Keq for dxr on growth rates (D) Effect of keff for the reaction RHCCE on proteome allocation (E) Effect of Keff and protein properties (unfolding Keq and aggregation propensities) for the reaction RHCCE on growth rates.

There is an alternative reaction CYSTL to homocysteine, encoded by either MetC or MalY. CYSTL (Cystathionine b-lyase)

$$cyst\_L\_c + h2o\_c \rightarrow hcys\_L\_c + nh4\_c + pyr\_c$$

Simulations indicate that, when Keff was increased from 0.01 to 28.8 for RHCCE, the mass fraction for S-ribosylhomocysteine cleavage enzyme was decreased from 0.02920 to 0.00001, but the flux through RHCCE was increased from 0.03157 to 0.03281 mmol g$^{-1}$ h$^{-1}$, and meanwhile the mass fraction for Cystathionine b-lyase was increased from 0.0000763 to 0.0000773 with a concurrent increase in the flux through CYSTL from 0.13087 to 0.13242 mmol g$^{-1}$ h$^{-1}$. Thus, CYSTL is the main source of homocysteine. A change in Keff for RHCCE, which is already small in EcoliME, would have an insignificant effect on the growth and metabolism of *E. coli*.

Fig 3E reveals that, in addition to the slight change of growth rates when a wide range of Keff values were applied to RHCCE, the change of protein properties (unfolding Keq and aggregation propensities) does not affect growth rate. This insensitivity further confirms that the RHCCE reaction, as other reactions with lower default Keffs, is not as important as the DXPRli reaction in the StressME model.

Collectively, the quality control pipeline identified a key Keff for the reaction DXPRli. If uncoordinated, it may cause overexpression of *dxr* and its corresponding chaperones under thermal stress. It may lead to an incorrect characterization of the stress response predicted by StressME for the heat-evolved strain.

It should be noted with interest that *in vitro* characterization of DXPRLi has shown that this enzyme can be heat-stable up to 60°C [21]. Therefore, this key enzyme, once adapted *in vivo* under higher temperature, may not need more proteome resources to synthesize itself and its protective chaperone, thereby saving more resources for other protein synthesis that can support growth under thermal stress. This may explain why the difference in DXPRli Keff can cause the heat-evolved strain to be distinct from the wild type strain.

## Mass balance check for metabolome and proteome

We further assessed the quality of the StressME model for all test conditions by checking the mass balance for metabolome and proteome. For all 1673 metabolites, we obtained a perfect match between the consumption and production of each metabolite (Fig A in S5 Appendix). For all 1578 proteins under all conditions, we consistently obtained the total simulated mass fraction (modeled_protein_fraction) very close to the setup value at 90% (Fig B in S5 Appendix), i.e., modeled_protein_fraction should be close to {1 –unmodeled_protein_fraction}, where unmodeled_protein_fraction is a model parameter set to 10% based on experimental quantification of the proteome mass (around 5% unmeasured) [22] and previous ME models (around 15% unmodeled) [12].

## Phenotype validation using consolidated kinetome

Finally, we consolidated a single kinetome that represents heat-evolved stress response (published FoldME results), while also correctly reproducing the published acid and oxidative ME-model stress responses. Based on results described above, this consolidation required only modifying one Keff relative to the AcidifyME/OxidizeME kinetome: DXPRli keff from 0.01 to 88.72 s$^{-1}$.

Using this consolidated Keff, we simulated growth rates for the wild-type and heat-evolved strains exposed to individual stresses (thermal, oxidative, acid). Fig 4 indicates that the final

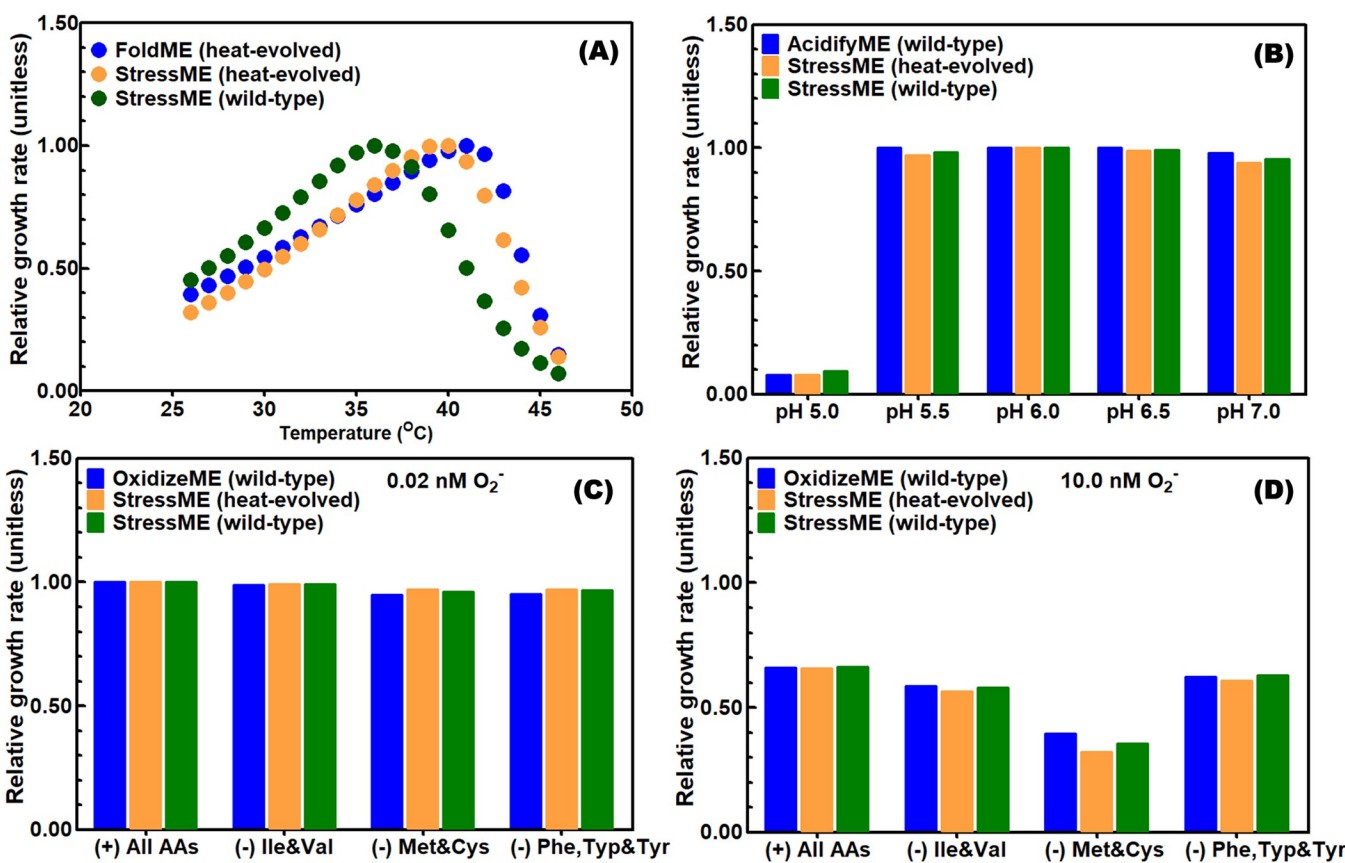

**Fig 4. StressME accurately simulates individual stress responses.** (A) thermal stress (B) acid stress (C) and (D) oxidative stress with different supply of amino acids (AAs).

StressME model with Keffs for the heat-evolved strain simulates stress responses that closely resemble those of single-stress ME models, including the strain growth rate dependence on temperature (Fig 4A), pH (Fig 4B), ROS levels (Fig 4C and 4D), and different supplementation of amino acids (AAs, see Fig 4D). Here the AA supply was used to alleviate the deactivation of AA biosynthesis pathways by ROS. It also shows that the StressME with Keffs for the wild-type strain can reproduce what have been experimentally observed under various stress conditions for this strain [10–12].

It should be noted that, by only coordinating a Keff value for one reaction (DXPRli) in the heat-evolved strain, the integrated StressME is able to reproduce phenotypes reported by all single Stress ME models. In addition, the StressME with the wild-type Keffs can correctly describe the phenotypes obtained by OxidizeME and AcidifyME, and those experimentally observed for the wild-type strain (e.g., the growth maximization at around 37˚C). Thus, it confirms the overall robustness of StressME–if more genes, proteins, metabolites, and reactions are added to the model, re-fitting the Keffs for all other reactions is likely not required.

Small differences in individual stress response simulations are likely due to: (1) mechanisms inherent with single stress have been integrated and therefore, there are some interconnections among individual stress-response mechanisms in StressME (2) unmodeled protein fractions have been changed due to new proteins added.

There are some differences in the exchange fluxes with temperatures between the wild type and the heat-evolved strains, as shown in Fig A in S6 Appendix. It indicates the different

temperature that the wild-type started decreasing the metabolic activity (around 36°C) from that of the heat-evolved (42°C). This difference confirms that the heat-evolved strain has adapted to the higher temperatures in terms of the metabolic activity.

## Alternative optima (strategies) to face stressors

**Flux variability analysis.** Flux variability analysis (FVA) was performed for the heat-evolved (Fig 5A–5D) and wild-type (Fig 5E–5H) StressME to find the minimum and maximum exchange fluxes by fixing 95% - 100% growth rates under different temperatures, while minimizing and maximizing the acetate production rates (APR), oxygen uptake rates (OUR), glucose uptake rates (GUR) and CO2 production rates (CPR).

For those observable phenotypes to support 95%-100% of maximal biomass production rates, FVA found the boundedness of the optimal solution space, with narrow ranges by fixing 100% growth rates. It suggests limited alternative optima that can be obtained by StressME, confirming the robustness of the model. Interestingly, the 100% growth FVA identified three stages of acetate metabolism over a wide range of temperature for both the wild-type and heat-evolved strains, i.e., an active acetate overflow from 24 to 30°C, and from 36–46°C, but an inactive acetate overflow between 31 and 35°C. At lower and higher temperature, the 100% growth FVA predicted a relatively wider solution space for APR in the wild-type than in the heat-evolved, suggesting that the acetate overflow may be an efficient way by which cells adjust their proteins in response to stress conditions.

StressME predicted the excretion of acetate at higher temperature for both wild-type and heat-evolved strains, as shown in Fig A in S6 Appendix, which is consistent with the experimental observation [23]. However, the acetate production rates were found to decrease after 42°C for both strains, which was accompanied by a decrease in glucose and oxygen uptake rates and a decrease in CO2 excretion rates, starting from 42°C for the heat-evolved strain but earlier at around 35°C for the wild type. This significant concurrent decrease was associated with the downregulation of the metabolic activities.

The acetate overflow metabolism was found to terminate between 30 and 36°C for both strains, but it resumed to an active state after 36°C. It suggests that cells may use the acetate overflow metabolism to switch from fully respiratory to respiro-fermentative growth to maintain their metabolic activities under stress.

**Alternative optima captured by StressME (purT vs. ackA).** It should be noted that some other alternative optima may be captured by StressME. Although they do not affect the phenotypes, they may affect the predicted local distribution of the proteome and fluxome. One example of such alternative optima is the presence of the three proteins (*TdcD*, *PurT* and *AckA*) with any one capable of catalyzing the reaction ACKr according to the gene-protein-reaction (GPR) rule in EcoliME [9]. We use the systematic analysis of the protein structure, stability, and function to confirm that the *purT* expression preferable to *ackA* at lower temperatures is due to the alternative optima (see S7 Appendix).

The proteome resource allocation at 26°C indicates a significant increase in the synthesis of *purT* protein (GAR transformylase-T), as shown in Fig A in S7 Appendix. GAR transformylase-T is mainly for catalyzing the purine and pyrimidine biosynthesis. However, it is also able to catalyze the cleavage of acetyl phosphate to produce acetate with ATP (reaction ACKr) [24]. The other dual-function enzyme that has similar catalytic mechanisms of GAR transformylase-T (*purT*) is propionate kinase (*tdcD*), which is mainly responsible for the conversion of propionyl phosphate and ADP to propionate and ATP but also possess acetate kinase activity (*ackA*).

The structure of three proteins (*tdcD*, *purT* and *ackA*) is shown in Table A in S7 Appendix, with any one of the three capable of catalyzing the reaction ACKr according to the gene-

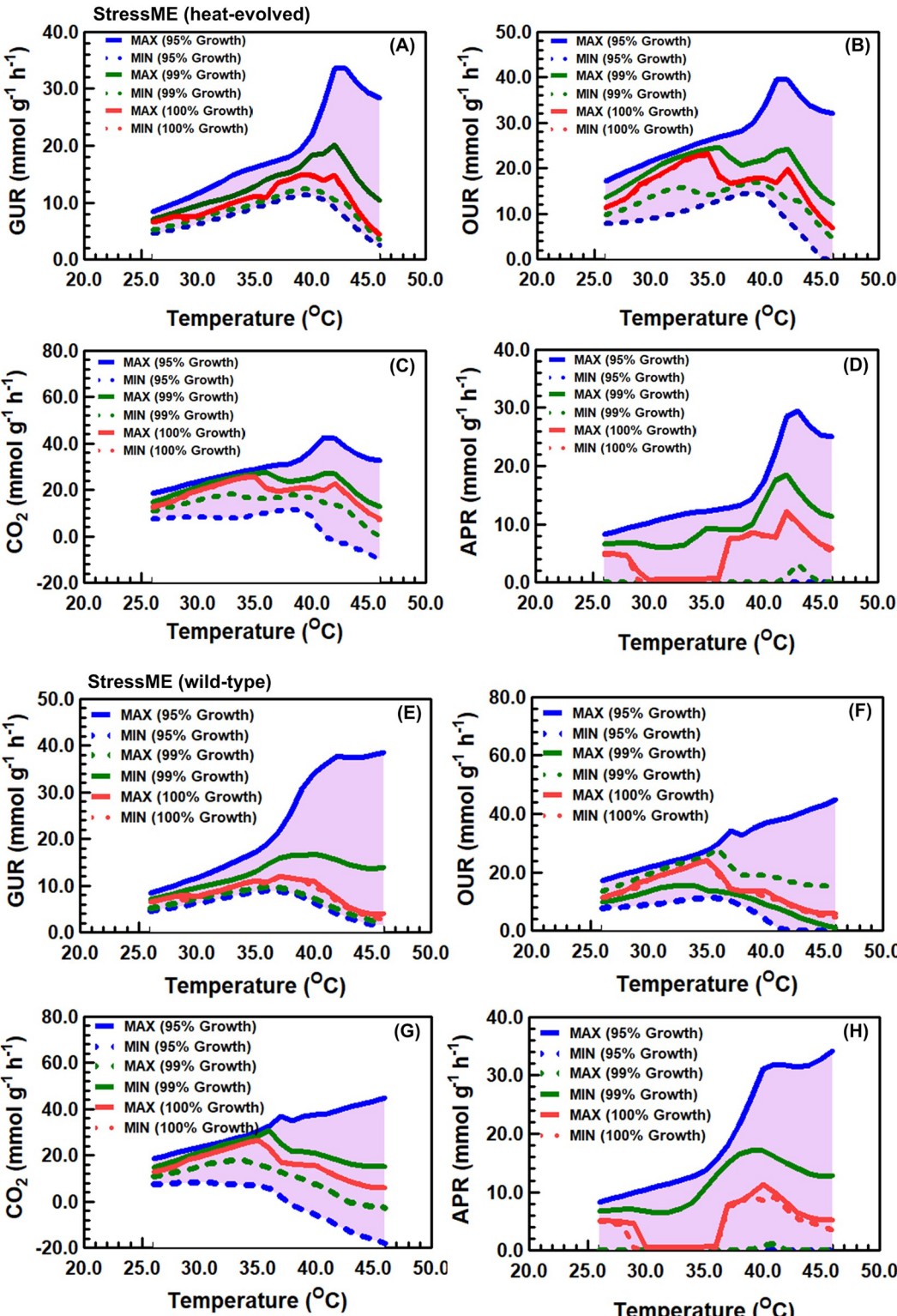

**Fig 5. Computing alternative optimal solutions from StressME by Flux variability analysis (FVA).** FVA for the heat-evolved strain (A)-(D): (A) glucose uptake rates (GUR), (B) oxygen uptake rates (OUR), (C) CO2 production/uptake rates (CO2) and (D) acetate production rates (APR). FVA for the wild-type strain (E)-(H). Pink zones represent the maximal boundedness of the optimal solution space to support 95%-100% of maximal biomass production rates. Negative values indicate the reverse direction of the exchange reaction.

protein-reaction (GPR) rule in EcoliME [9]. These three proteins are similar to each other in structure such as molecular weight and number of residues, suggesting that the cost for protein biosynthesis is not the reason for StressME to prefer *purT* protein to *tdcD* and *ackA* proteins.

The effect of protein stability on the synthesis preference was examined by changing up to 100-fold of equilibrium constant of unfolding (Keq), aggregation propensities (agg) and protein kinetic folding rate [kf] for *purT* and *ackA* proteins, respectively. The StressME simulations indicate that the proteome, as well as the phenotypes and fluxome, is not significantly affected by the protein stability, thus ruling out other possible stability causes determining the choice of purT for acetate kinase activity.

Finally, the mRNA translation and protein synthesis by *purT* gene was turned off in StressME at 26˚C. The proteome clearly shows that, when *purT* was turned off, the same amount of acetate kinase (*ackA*) was synthesized at 26˚C. However, the synthesis of acetate kinase was blocked when *purT* was turned on again (Fig A in S7 Appendix). Fig A in S7 Appendix further indicates that the acetate overflow metabolism observed at 26˚C but almost disappeared at 32˚C can also be caused by acetate kinase (*ackA*) if the purT protein synthesis is blocked. These simulations suggest that the *purT* expression preferable to *ackA* at 26˚C is due to the fact of the alternative optima captured by StressME, rather than the 'hidden' biological mechanism that may significantly affect the overall proteome and fluxome.

## Investigating cross-stress and cross-talk resistance using StressME

Cross-stress resistance is a complex systems-level phenomenon that continues to be studied actively, and thus presents a good case study for StressME's utility. Cross-stress resistance in microbes refers to the acquisition of resistance to a second type of stress, after exposure to a different primary stress [25]. Understanding of cross-stress resistance has advanced from being attributed to universal mechanisms (e.g., general stress response), to specific mechanisms that depend strictly on the primary stress [25]. Thus, the type and order of applied stresses determines cross-stress resistance: e.g., heat tolerance acquisition after acid [26,27] or oxidative stress in various organisms [25]. Intuitively, certain stress-induced factors have protective activities under multiple stresses–e.g., trehalose was hypothesized to protect membrane by stabilizing polar groups of phospholipids, scavenges free radicals, and is an energy reservoir [25]. However, later studies refuted the role of trehalose as providing general stress protection and instead attributed greater importance to its specific role for energy metabolism, or with altered proteome expression associated with it [25]. Furthermore, studies appreciate the role of repressing large sets of genes (e.g., growth processes) to enable the expression of other proteins important for stress response [25]. However, the exact role of each expressed gene for cross-stress response is difficult to pinpoint.

Cross-talk resistance, on the other hand, is another complex systems-level phenomenon that integrates different stress-response pathways under multiple stresses to maximize the protective performance. An enhanced understanding of the cross-talk mechanisms at the system level will have potential practical implications, such as in food industrial processes, microbial waste treatment and high-density cultivation for microbial products, where cells may have a higher chance of experiencing a set of environmental stressors simultaneously.

With these perspectives, we investigate both system-level proteome allocation, and specific biochemical activities associated with cross-stress and cross-talk mechanisms. We first consider two scenarios:

- Scenario A: Exposure to Stress 1 → no long-term adaptation (wildtype) → exposure to Stress 2

- Scenario B: Exposure to Stress 1 → adaptation to Stress 1 (heat-evolved) → exposure to Stress 2

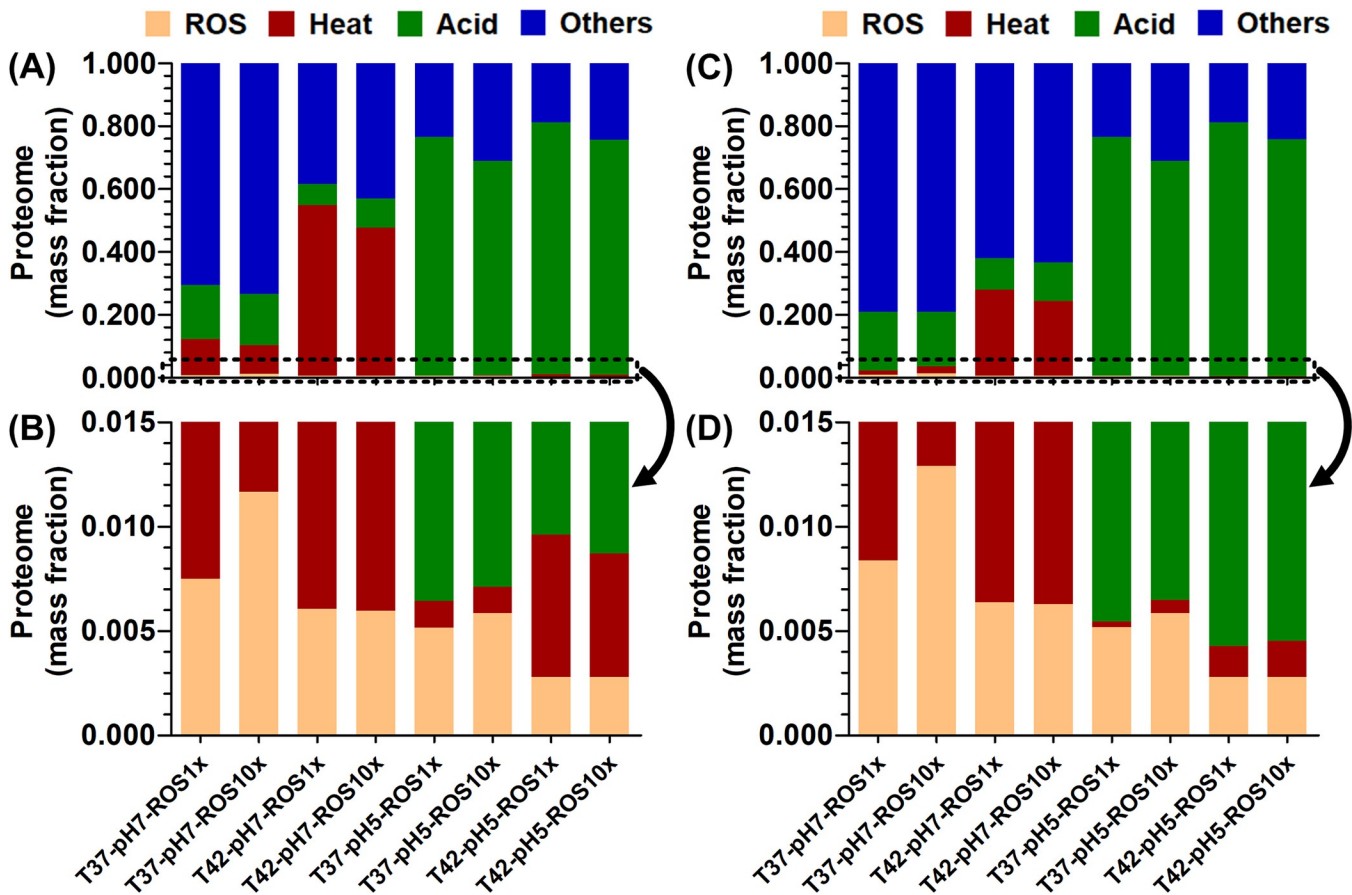

**Fig 6. Effect of heat (Stress 1) adaptation on system-level proteome re-allocation as tradeoffs when exposed to Stress 2 (thermal or acid or oxidative or combination).** (A) wild type strain–proteome mass fraction overview from 0.0 to 1.0 (B) wild type strain–proteome zoom-in view from 0.000 to 0.015 (C) heat-evolved strain–proteome mass fraction overview from 0.0 to 1.0 (D) heat-evolved strain–proteome zoom-in view from 0.000 to 0.015.

For Scenario B, we assume the cell has adapted to Stress 1 on an evolutionary timescale.

In our *in-silico* experiments, we investigate how a heat-adapted or non-adapted strain (each having different Keffs) responds to individual stresses (heat, ROS, acid) or combination of these stresses (i.e., thermal-oxidative, thermal-acid, oxidative-acid, and thermal-oxidative-acid) simultaneously.

We first compared proteome allocation between heat-adapted (heat-evolved) and heat non-adapted (wild-type) strains toward ROS, heat, and acid-response across all three stresses: ROS from 1x to 10x, temperatures from 37 to 42˚C, and pH 7.0 and 5.0. These simulations were repeated for wild-type and heat-evolved Keff vectors.

We found an increased stress resistance for the heat-evolved strain (Fig 6C and 6D) when subjected to stress 2 with respect to single (thermal) or dual (thermal-oxidative) stress. The thermal-activated proteome mass fraction in the heat evolved strain could decrease to 23.8% from 47.1% in the wild type under 10x ROS and 42˚C heat stresses (Fig 6A and 6B). Thus, heat-adaptation may synergistically improve ROS response by freeing up proteome resources that would otherwise be allocated to chaperones. The ROS-activated proteome mass fraction increased by 5.2% for the heat-evolved strain over wild-type under 10x ROS and 42˚C heat stresses. Meanwhile, the heat evolved strain showed more active metabolic activities (growth rate 0.82 h$^{-1}$) than the wild type strain (growth rate 0.32 h$^{-1}$).

When subjected to acid stress, the heat-evolved strain could not remarkably improve performance under single, dual, and triple stresses. The acid-activated proteome only increased up to 0.80% for heat-evolved vs. wild-type simulations under heat (42˚C) and acid stresses (pH 5.0). Meanwhile, both strains showed similar low metabolic activities (i.e., growth rate 0.071 ~ 0.087 h$^{-1}$ at 37˚C and 0.031 ~ 0.035 h$^{-1}$ at 42˚C).

Hence, acid response is not likely to benefit by proteome allocation-based mechanisms arising from the exposure to stress 1, since the largest contributors to acid response in our model are the periplasmic chaperones (*Hde*) that do not compete directly with cytoplasmic proteins. Therefore, we posit that cross-stress protection can be explained through proteome allocation trade-offs. Furthermore, these trade-offs require precise characterization as they depend on the identity of proteins activated and their localization, which will be discussed in the next sections.

**Heat adaptation can induce cross-protection against thermal or thermal-oxidative stress.**   First, under simultaneous thermal-oxidative stress, all individual stress responses are activated. Wild-type simulations indicate the mass fractions for the COG functional O-category (posttranslational modification, protein turnover, chaperones) increased from 15% (37˚C) to 63% (42˚C) at low ROS levels (0.01x) (Fig 7C). Increasing ROS to 10x lowered O-category expression moderately: 11% (37˚C) to 53% (42˚C). Thus, chaperone expression increases strongly with heat stress, as expected. Meanwhile, heat-adapted simulations show the O-category mass fractions only increasing from 10% (37˚C) to 27% (42˚C) when increasing ROS to 10x (Fig 7A). Thus, ROS response may benefit from the proteome re-allocation due to the heat adaptation.

Fig 7B and 7D indicate the ROS-activated proteome (19 ROS stress-exclusive and stress-intensified proteins) [12] and ROS-vulnerable proteome (31 [Fe-S] binding proteins) [12] for the heat-adapted and wild-type strains, respectively. The ROS-activated and ROS-vulnerable proteome mass fractions increased by 4.7% and 8.9% for the heat-evolved strain over the wild-type under 10x ROS and 42˚C heat stresses. Among ROS-vulnerable [Fe-S] binding proteins, quinolinate synthase (*nadA*) had a significant increase (10.4%) in the mass fraction for heat-evolved vs. wild-type simulations. The damage fluxes for *nadA* due to ROS were increased from 1.27 and 5.11 nmol g$^{-1}$ h$^{-1}$ to 96.8 and 387 nmol g$^{-1}$ h$^{-1}$, respectively, when ROS rose from 0.01x to 10x basal level. To complement such ROS damage loss, the expression for *nadA* should increase if the proteome availability is sufficient to allow such a re-allocation.

Another important ROS-vulnerable [Fe-S] binding protein is NADH dehydrogenase encoded by *nuo* (S8 Appendix). The *nuo* proteome (13 proteins) mass fractions increased by 7.5% for the heat-evolved strain over the wild type under 1x ROS and 37˚C. As ROS increased to 10x, nuo proteome expression terminated for both strains, but an alternative NADH dehydrogenase encoded by *ndh* (S8 Appendix) significantly increased, suggesting that cells tried to avoid the uneconomical *nuo*-coded NADH pathway vulnerable to the ROS damage. The heat-adapted strain still showed 12.2% more in *ndh* expression than the wild-type after the NADH pathway shift. Under ROS and 42˚C stresses, both strains terminated the *nuo* and *ndh* expression, suggesting that there is no sufficient proteome availability for ROS response to be more comprehensive.

**Heat adaptation cannot induce cross-protection against acid-involved stress.**   Fig 8 shows the proteome reallocation under thermal-acid stress for the heat-evolved and wide-type strain. The heat adaptation released some cytosolic proteome resources that would otherwise be allocated to chaperones. Thus, the heat-evolved strain caused more remarkable cytosolic proteome reallocation than the wild type under acid stress from low to high temperature. However, the acid stress increased requirement for acid response proteins such as periplasmic chaperones (*HdeB*), periplasmic membrane synthesis (*LptA*) and sigma transcription factor

## Heat adapted strain

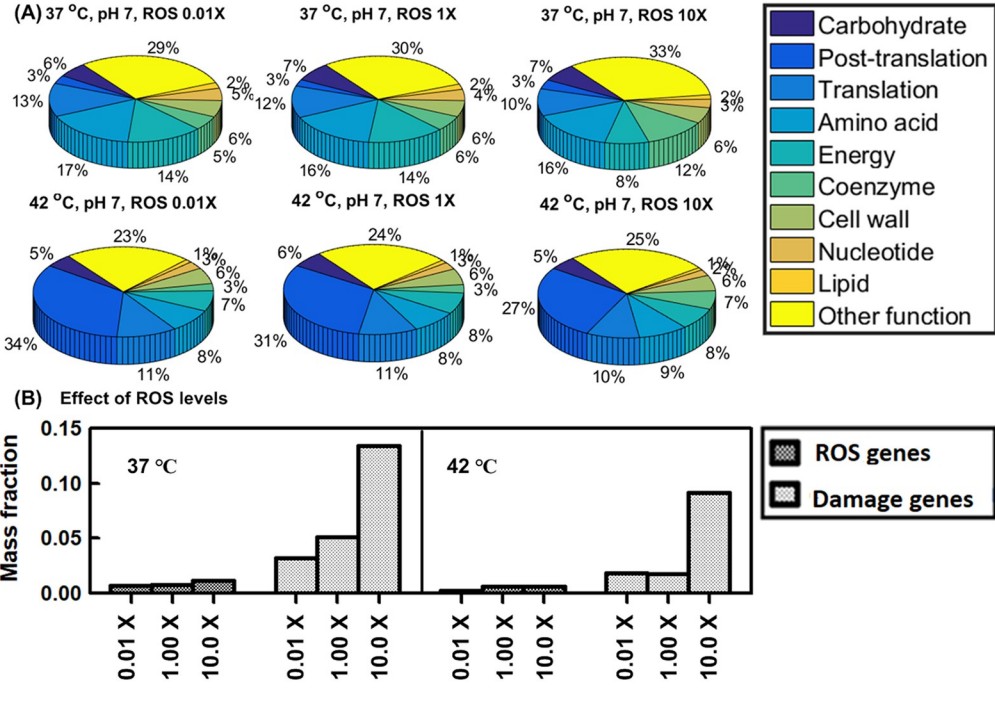

## Heat non-adapted strain

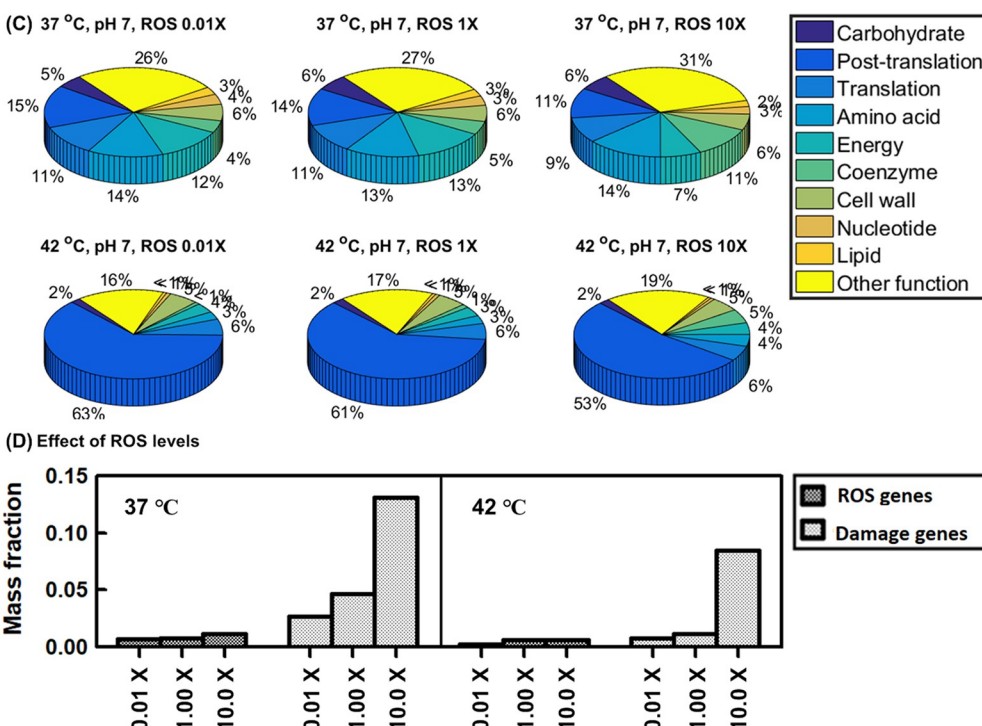

**Fig 7. Proteome reallocation under thermal and oxidative dual stress.** Heat-evolved strain (A)-(B): (A) proteome mass fractions for COG functional groups (B) proteome mass fractions for key oxidative-response groups. Wild-type strain (C)-(D). ROS genes: ROS-activated proteome (19 ROS stress-exclusive and stress-intensified proteins). Damage genes: ROS-vulnerable proteome (31 [Fe-S] binding proteins).

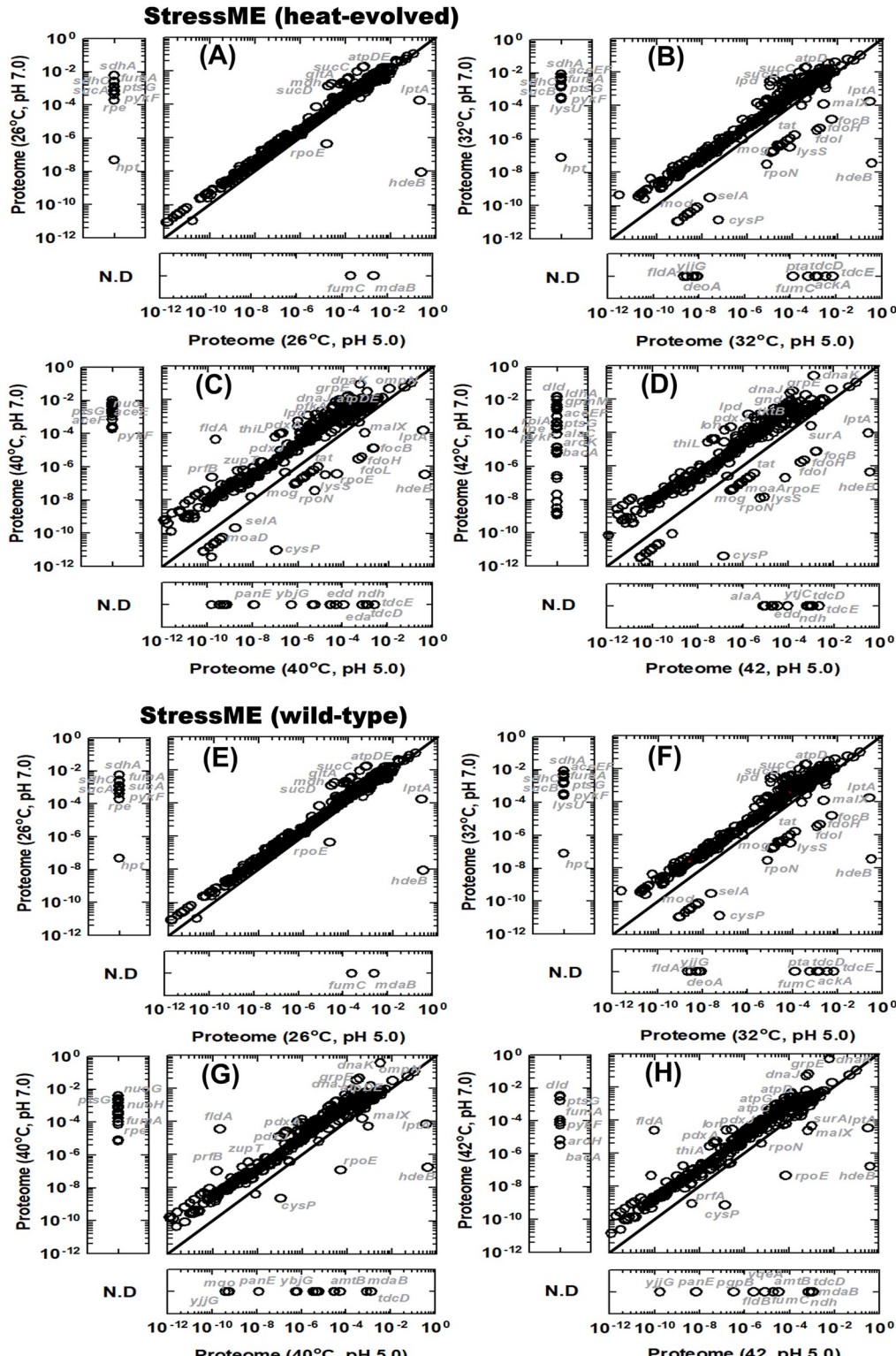

**Fig 8. Proteome reallocation under thermal-acid dual stress.** Heat-evolved strain (A)-(D): (A) pH 5.0 and pH 7.0 at 26˚C; (B) pH 5.0 and pH 7.0 at 32˚C; (C) pH 5.0 and pH 7.0 at 40˚C and (D) pH 5.0 and pH 7.0 at 42˚C. Wild-type strain: (E)-(H).

(*rpoE*) for extracytoplasmic activities [28]. These periplasmic proteins do not compete directly with cytoplasmic proteins, which may reduce the benefit by proteome re-allocation due to the heat adaptation.

The metabolic flux balance analysis (MFBA) further confirmed that the heat adaptation could not induce cross protection against acid-involved stresses. Both strains showed similar intra- and extra-cellular flux distribution under acid stress (pH 5.0) (Fig 9B and 9D), although the heat-evolved strain indicated a more active metabolic status than the wild type at pH 7.0 (Fig 9A and 9C).

Under acid stress, the MFBA captured a potential switch in electron transfer pathways from succinate oxidation via succinate dehydrogenase (*sdh*) to formate oxidation by formate dehydrogenase (*fdo*) in the heat-evolved strain (Fig 9B). This shift requires pyruvate formate lyase (*tdcE*), which is inactive at pH7 according to StressME simulations for the heat-evolved strain (Fig 9A). Thus, the StressME simulations suggest that the heat-evolved strain may change the metabolism under acid stress from fully respiratory through TCA to respiro-fermentative metabolism through anaerobic glycolysis. The respiro-fermentative metabolism has been reported for *E.coli* under both aerobic [29,30] and anaerobic conditions [31,32]. However, there has been no work on the effect of the pH on the *E.coli* heat-evolved strain with respect to the respiro-fermentation. The possible reason for this shift is because the *E.coli* heat-evolved strain may use the proteome-ready-to-deploy respiro-fermentative pathways [33], where the pyruvate formate lyase is constitutively expressed in *E.coli* [34], to avoid the proteome limitation for energy biogenesis under acid stress. It has also been reported that the proteome cost for energy biogenesis is less expensive by fermentation than by respiration in *E.coli* [35], justifying the choice of the respiro-fermentative metabolism under proteome limitation. The simulations indicate that this switch is an inherent characteristic of the wild type because it was active even under single thermal stress (Fig 9C). The wild type may use this respiro-fermentative metabolism to avoid the proteome limitation under thermal stress, a mechanism that the heat evolved strain does not need because it has already adapted to the thermal stress. Further experimental work is required to validate the simulated switch between TCA and respiro-fermentative metabolism under different stress conditions.

The less use of the TCA cycle can explain the mechanism of acetate overflow under single and cross-stress conditions. When the TCA activities were downregulated in both strains, the acetyl-CoA partially going to TCA under aerobic conditions was converted to acetate by the PTA-PCKA pathway, causing the phenotype of the acetate overflow.

We further studied proteome allocation between the heat-adapted and heat non-adapted strains toward oxidative-acid dual stress and thermal-oxidative-acid triple stress. We confirmed that heat adaptation cannot induce cross protection against the acid-involved cross stresses, as shown in the similar mass fractions for the COG functional categories and ROS related proteins between these two strains (Figs 10 and 11).

Hence, the StressME may be used as an *in-silico* platform to develop strategies for experimental evolution of super bacteria to enhance the cross-stress resistance or optimal control of the extracellular environment to maximize the cross-talk protection (e.g., to get rid of the acidity from multiple stressors first).

## Materials and methods

### Software

This work used the EcoliME framework developed in CobraME [9] to build an integrated StressME model from three published single stress ME models–FoldME [10], AcidifyME [11] and OxidizeME [12].

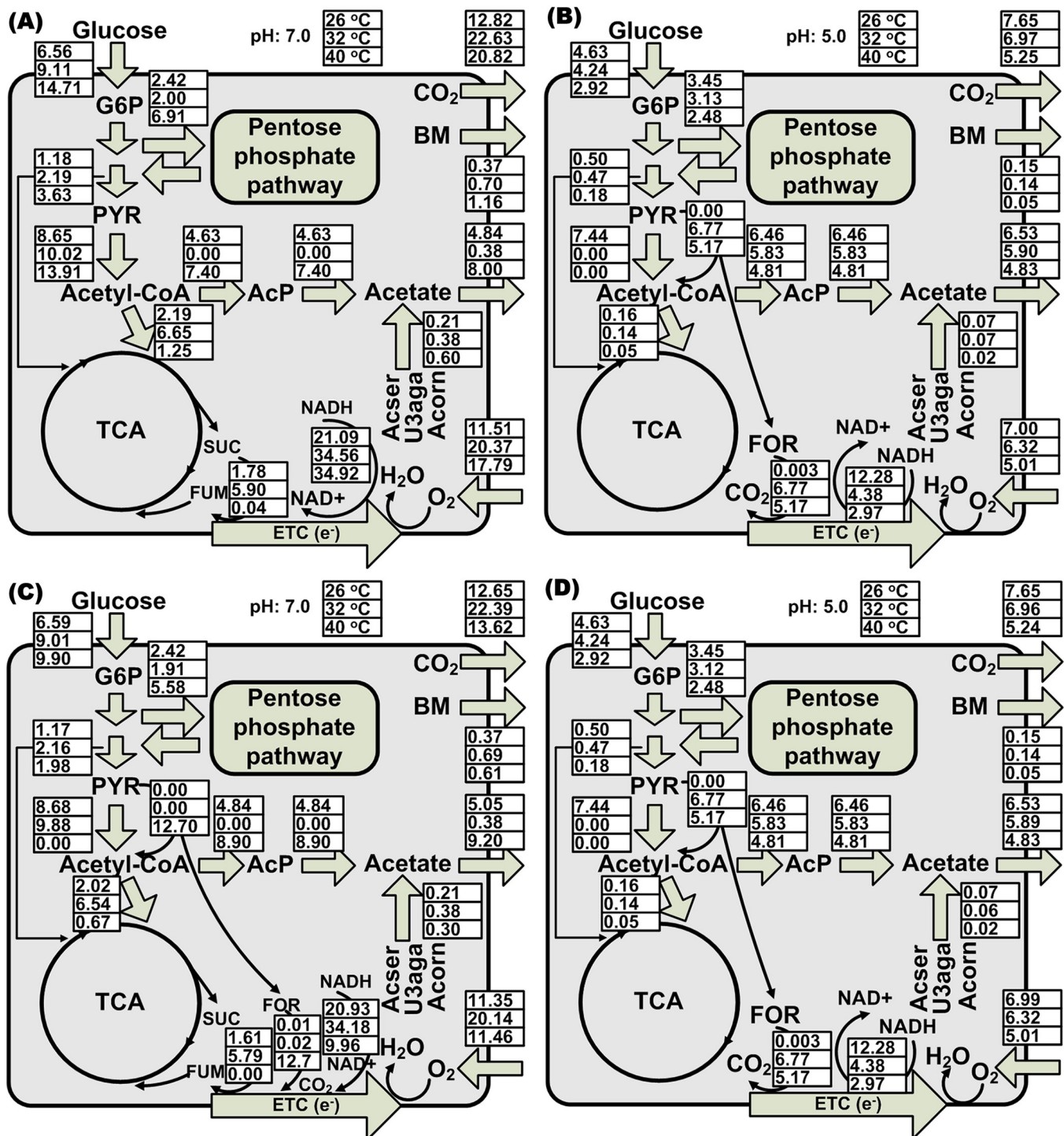

**Fig 9. Metabolic flux balance analysis (MFBA) showing a switch from fully respiratory to respiro-fermentative metabolism and acetate overflow under thermal-acid stress**: Heat-evolved strain (A) and (B); Wild-type strain (C) and (D). (A) and (C): Fluxome (mmol g$^{-1}$ DCW h$^{-1}$) at pH 7.0, temperature 26°C, 32°C and 40°C. (B) and (D): Fluxome at pH 5.0, temperature 26°C, 32°C and 40°C.

All code was tested in Python version 3.6.3, which is the default version for our StressME package. We converted any Python v2.7 scripts of previously published single StressME codebases to Python v3.6. Three updated ME model packages can be downloaded from https://

## Heat adapted strain

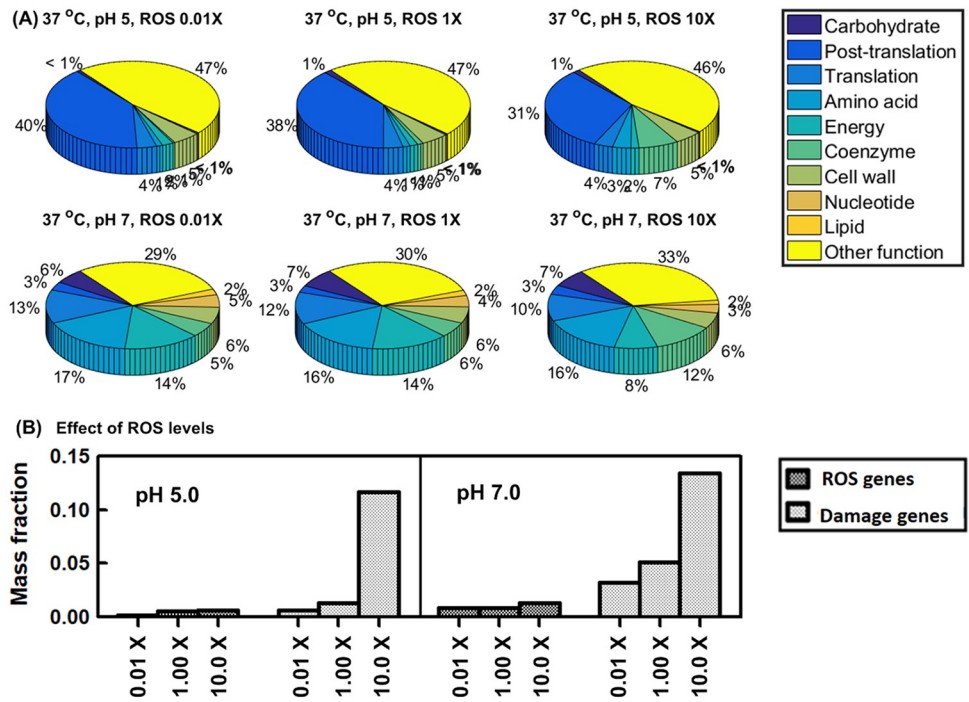

## Heat non-adapted strain

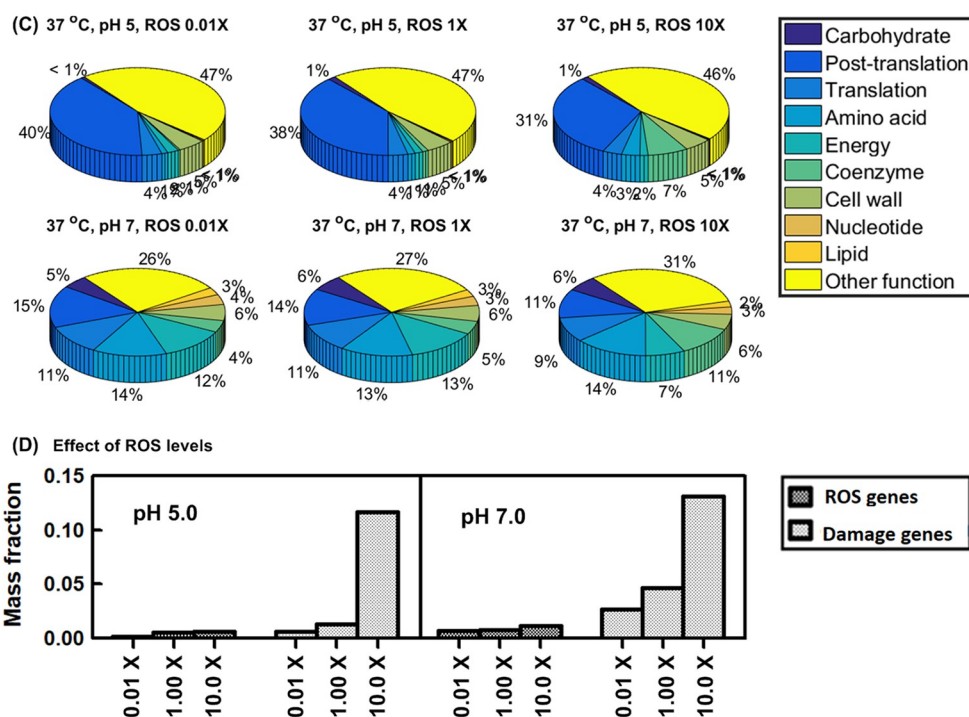

**Fig 10. Proteome reallocation under oxidative and acid dual stress.** Heat-evolved strain (A)-(B): (A) proteome mass fractions for COG functional groups (B) proteome mass fractions for key oxidative-response groups. Wild-type strain (C)-(D). ROS genes: ROS-activated proteome (19 ROS stress-exclusive and stress-intensified proteins). Damage genes: ROS-vulnerable proteome (31 [Fe-S] binding proteins).

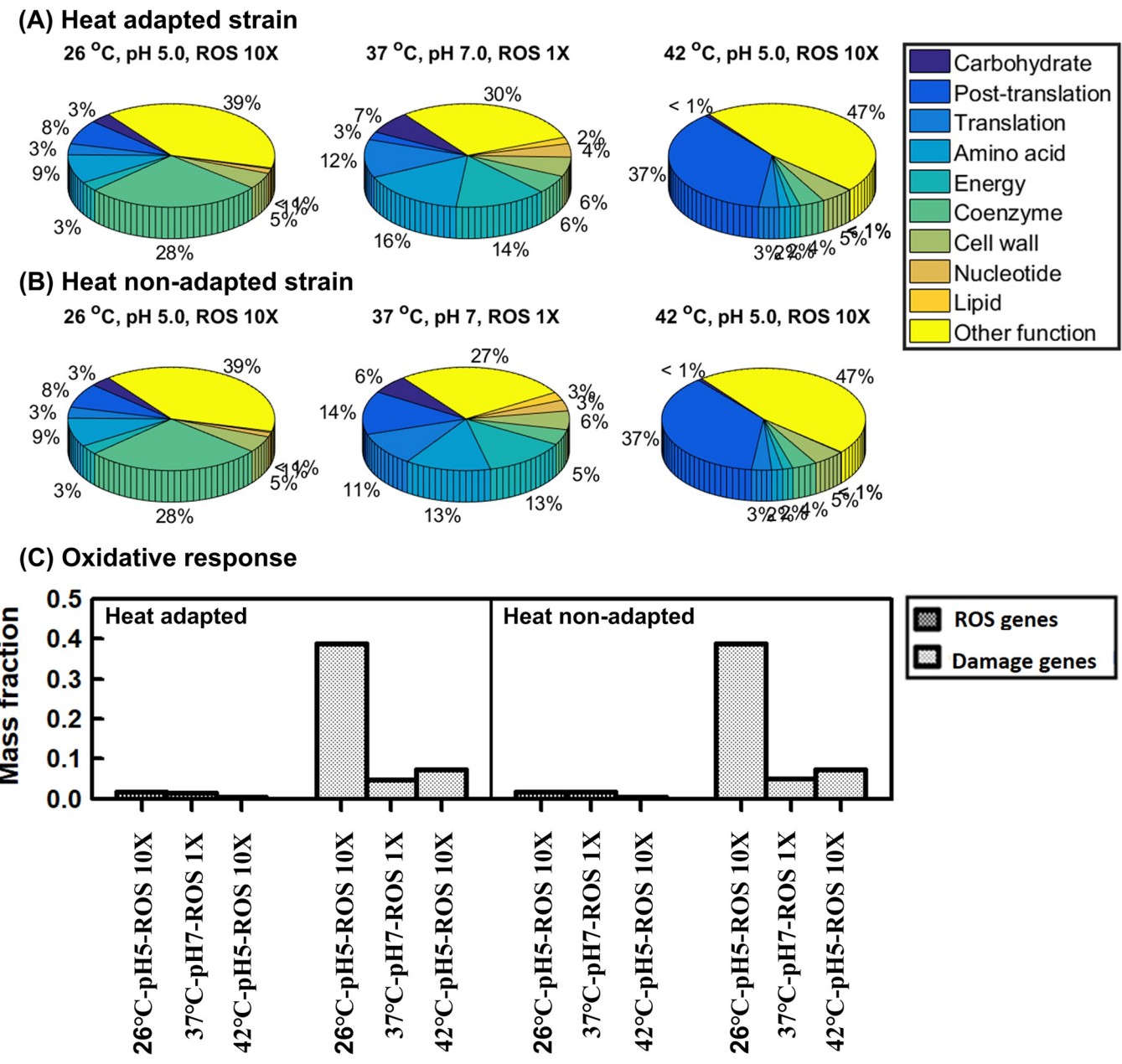

**Fig 11. Proteome allocation under thermal-oxidative-acid triple stress.** (A) proteome mass fractions for COG functional groups in heat-adapted strain (B) proteome mass fractions for COG functional groups in heat non-adapted strain (C) ROS-activated proteome (19 ROS stress-exclusive and stress-intensified proteins) and ROS-vulnerable proteome (31 [Fe-S] binding proteins) for heat-adapted and heat non-adapted strains.

github.com/QCSB/StressME. We also provide supporting scripts for COBRAme (model building and simulation scripts) that we modified for compatibility with Python 3.6.3 and StressME (see S9 Appendix).

Experienced users can use instructions in S9 Appendix and three updated single stress ME models (https://github.com/QCSB/StressME) to build and extend StressME and run simulations in Linux clusters (S9 Appendix for details). All simulations in this study were performed in Compute Canada clusters based on this setup. The standalone executable StressME package

for running in personal computers is also provided by a docker container (queensysbio/
stressme:v1.1, see S10 Appendix for details).

## Integration of single stress models

The StressME was built and optimized by a pipeline to combine mechanisms of three single
stress ME models (thermal, oxidative, acid) in the EcoliME framework. First, three single stress
ME models (FoldME [10], OxidizeME [12] and AcidifyME [11]) were rebuilt from the same
starting point (E. coli K-12 MG1655 ME-model iJL1678-ME [9]) under Python version 3.6.3
based on the published protocols [9]. Second, eleven stress-response genes/proteins that were
not included in iJL1678-ME were identified. Among them, nine cytoplasmic proteins in
response to oxidative stress and one (ATP-dependent protease, Lon) associated with misfolded
protein degradation were added to the FoldME by setting up their temperature-dependent
protein kinetic folding rates Kf(T), thermostability ΔG(T), equilibrium constants of unfolding
Keq(T) and aggregation propensity (agg) (S3 Appendix). Thus, the StressME at this stage can
apply the protein folding network in cytoplasm (i.e., the competitive pathways of spontaneous
folding, the *DnaK*-assisted folding, and the *GroEL/ES*-mediated folding) to these added pro-
teins. Third, one periplasmic protein (*HdeB*) in response to acid stress was coupled to
StressME through AcidifyME describing the stability of periplasmic proteins as a function of
pH and temperature, and the protection of *HdeB* on unfolded periplasmic proteins through
spontaneous folding. At this stage, the change of *E. coli* membrane lipid fatty acid composition
and membrane protein activity with pH were also coupled to StressME through AcidifyME to
characterize the observed acid stress response. Finally, substances and reactions associated
with damage by reactive oxidative species (ROS) to macromolecules were coupled to StressME
through OxidizeME to reconstruct a final integrated StressME, including relevant mechanisms
of demetallation of Fe(II) proteins by ROS, mismetallation by alternative metal ions, oxidiza-
tion and repair of Iron–sulfur clusters, unincorporated Fe(II) reaction with $H_2O_2$ (Featon
reaction) to generate hydroxyl radicals, and the protection from Dps protein to store unincor-
porated Fe(II). The eleven proteins added to StressME are shown in Table 1. The coupling
scripts can be found in https://github.com/QCSB/StressME

The StressME was further wrapped in a user-friendly I/O platform to run simulations with
a simple input array of temperature, pH and ROS levels. The StressME simulations can auto-
matically generate CSV output for phenotypes, proteome and fluxome under all stress condi-
tions for further processing and visualization (Fig 12).

**Table 1. Eleven stress-response genes added to FoldME.**

| Gene | Symbol | Function | Stress |
|------|--------|----------|--------|
| b0605 | ahpC | reduction of hydroperoxide substrate. | Oxidative |
| b0606 | ahpF | reduction of hydroperoxide substrate. | Oxidative |
| b2962 | yggX | protect iron-sulfur proteins | Oxidative |
| b4209 | ytfE | Iron-sulfur cluster repair | Oxidative |
| b3662 | nepI | purine ribonucleoside exporter | Oxidative |
| b0812 | dps | DNA damage protection and iron sequestration | Oxidative |
| b3961 | oxyR | Regulator for the expression of antioxidant genes | Oxidative |
| b4062 | soxS | Transcriptional activator for the superoxide response regulon | Oxidative |
| b4063 | soxR | Redox-sensitive transcriptional activator of soxS | Oxidative |
| b0439 | lon | Degradation of abnormal proteins | Thermal |
| b3509 | hdeB | Periplasmic chaperone for acid stress protection | Acid |

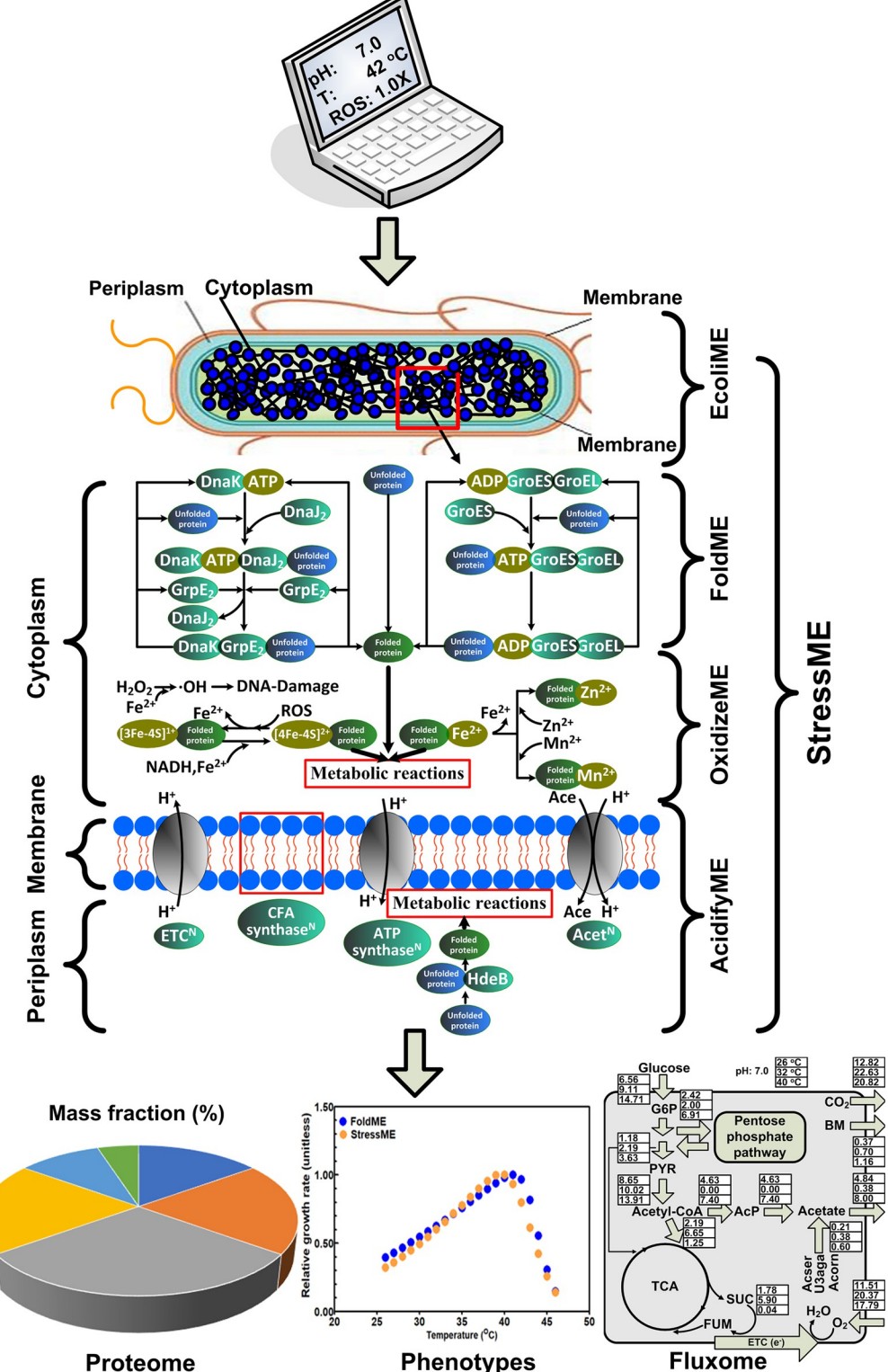

**Fig 12. Integrated StressME model to combine mechanisms of three single stress models (thermal, oxidative, acid).**
User-friendly I/O platform to run simulations with a simple array of temperature, pH and ROS levels; CSV output for phenotypes, proteome and fluxome for further visualization.

## Keff coordination and optimization for different strains

In StressME, the synthesis rate of the catalyzing enzyme for a metabolic reaction can be expressed as:

$$V_{\text{translation}} = \mu/\text{k}_{\text{eff}} * V_{\text{metabolic\_reaction}} \tag{Eq1}$$

Where μ/keff is the small amount of the catalyzing enzyme required for the metabolic reaction to carry a flux. Eq 1 indicates that if a Keff value is increased, the amount of the protein to be synthesized is decreased. There are uncertainties of Keffs for metabolic reactions. As a first step for the model quality control, two single Keff vectors were optimized for the StressME representing the wild-type and heat-evolved strains, respectively. This quality control is based on a protocol to change a minimal number of Keffs that together satisfy the different strain and stress conditions. This was implemented by computing growth rates when each metabolic reaction Keff of total 5266 metabolic reactions was altered at one time from default (wild-type) to alternative (heat-evolved). All simulated 5266 growth rates were then sorted, and the most influential reactions subject to Keff change were identified. The screening was performed in parallel in Compute-Canada clusters.

## Validation

Using the optimal Keff vectors, growth rates reported by three single stress ME models were validated by StressME for the wild-type and heat-evolved strains exposed to various single stress, i.e., temperature from 26˚C to 46˚C, pH from 5.0 to 7.0, and superoxide from 0.02 nM to 10 nM. For oxidative stress validation, different supply of amino acids (AAs) was used—full AAs, AAs without Ile & Val, AAs without Met & Cys and AAs without Phe, Typ & Tyr. The growth rates obtained by StressME were then compared with those published by FoldME, AcidifyME and OxidizeME, respectively, to check the model robustness and consistency after different mechanisms in response to different types of stress had been coupled to each other.

## Flux variability analysis

Flux variability analysis (FVA) was performed for the wild-type and heat-evolved StressME to determine the range of possible solutions for exchange fluxes bounded by maxima and minima. It was done by fixing 95% - 100% growth rates under different temperatures, while minimizing and maximizing the acetate production rates (APR), oxygen uptake rates (OUR), glucose uptake rates (GUR) and carbon dioxide rates ($CO_2$), respectively.

## Cellular response to multiple stressors

Proteomics and fluxomics analysis were done to study the cross-stress & cross-talk resistance for the wild-type (heat non-adapted) and heat-evolved *E. coli* exposed to dual (thermal & oxidative, thermal & acid and oxidative & acid) and triple stress conditions (thermal & acid & oxidative), respectively. The thermal & oxidative stress conditions were simulated by StressME to increase the temperature from 37˚C to 42˚C and the ROS from 0.01x to 10x basal level. The thermal & acid stress conditions were simulated by increasing the temperature from 26˚C to 42˚C and pH from 5.0 to 7.0. The oxidative & acid stress conditions were simulated by StressME to increase ROS from 0.01x to 10x and pH from 5.0 to 7.0. The triple stress conditions were simulated by increasing temperature from 26˚C to 42˚C, ROS from 1x to 10x, and pH from 5.0 to 7.0. The proteome reallocation in response to stressors was quantified by protein mass fraction, which is defined as a ratio of the individual protein synthesized to the total protein synthesized. The algorithm to compute the protein mass fractions was described

elsewhere [36], which is based on the molecular weight of the individual protein, the translation flux of the protein and the growth rate. The fluxomic reallocation due to cross-stress & cross-talk adaptation was evaluated by the intracellular metabolic fluxes (mmol g$^{-1}$ h$^{-1}$), which were computed by the quad-precision linear and nonlinear programming solver qMINOS 5.6 to maximize the growth rate under stress conditions. The focus was put on the central metabolism such as glycolysis, citric cycle, pentose phosphate pathways, and electron transport chain reactions.

## Supporting information

**S1 Appendix. Biological expansion of StressME.**
(DOCX)

**S2 Appendix. Detailed information about the reactions added to StressME at each reconstruction step.**
(XLSX)

**S3 Appendix. Temperature-dependent thermostability, aggregation propensity, and folding rate constants for ten folding proteins added to StressME.**
(DOCX)

**S4 Appendix. Typical computation time for running StressME.**
(DOCX)

**S5 Appendix. Material balance check for metabolites and proteome.**
(DOCX)

**S6 Appendix. Temperature-dependent phenotypes and proteome.**
(DOCX)

**S7 Appendix. Alternative optima captured by StressME (purT vs. ackA).**
(DOCX)

**S8 Appendix. Reactions for NADH dehydrogenase and Quinolinate synthase.**
(DOCX)

**S9 Appendix. StressME using Linux (clusters).**
(DOCX)

**S10 Appendix. StressME using docker.**
(DOCX)

## Acknowledgments

We thank Herbert Yao for the valuable suggestions on the paper.

## Author Contributions

**Conceptualization:** Jiao Zhao, Ke Chen, Bernhard O. Palsson, Laurence Yang.

**Data curation:** Jiao Zhao, Ke Chen, Laurence Yang.

**Formal analysis:** Jiao Zhao, Ke Chen, Bernhard O. Palsson, Laurence Yang.

**Funding acquisition:** Bernhard O. Palsson, Laurence Yang.

**Investigation:** Jiao Zhao.

**Methodology:** Jiao Zhao, Ke Chen, Laurence Yang.

**Project administration:** Bernhard O. Palsson, Laurence Yang.

**Resources:** Laurence Yang.

**Software:** Jiao Zhao.

**Supervision:** Laurence Yang.

**Visualization:** Jiao Zhao.

**Writing – original draft:** Jiao Zhao.

**Writing – review & editing:** Jiao Zhao, Ke Chen, Bernhard O. Palsson, Laurence Yang.

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
