## [Decision Letter · Decision Letter 0]

31 Oct 2023

Dear Dr. Yang,

Thank you very much for submitting your manuscript "StressME: unified computing framework of Escherichia coli metabolism, gene expression, and stress responses" for consideration at PLOS Computational Biology. As with all papers reviewed by the journal, your manuscript was reviewed by members of the editorial board and by several independent reviewers. The reviewers appreciated the potential of generating an ME model that can respond to multiple stress conditions with improved prediction accuracy, while successfully reproducing results from previously reported single-stress models . Based on the reviews, we are likely to accept this manuscript for publication, providing that you modify the manuscript according to the review recommendations.

An important point brought up by reviewer #1 indicates that the link to the github repository is broken (generates error 404). Correcting this is essential for readers and reviewers to access the code. Make sure that the repository has an adequate level of documentation for users to download, install and test the scripts.

Sincerely,

Arturo Medrano-Soto, Ph.D.

Guest Editor

PLOS Computational Biology

Mark Alber

Section Editor

PLOS Computational Biology

Reviewer's Responses to Questions

**Comments to the Authors:**

Reviewer #1: In this work, three individual stress ME models of E. coli were integrated into one, which was then used to simulate the cellular metabolism and protein allocation in response to various stress conditions and their combinations. I think this work could benefit the understanding of how industrial microorganisms respond to challenging conditions at the cellular level. I have several questions and comments:

1. In Introduction, the authors should enumerate specific scenarios where StressME could find applications. While it seems that information was provided in the last paragraph of the manuscript, it should be included in the Introduction to better justify the development of this tool.

2. In the Quality control of the StressME model section, according to the authors, the parameters of the three individual ME models were obtained by fitting with experimental data, while the parameters of StressME were directly adapted from those models except the Keff of DXPRli. I wonder if the StressME parameters can be estimated in the same way.

3. In the Keff in single- and Integrated-StressME models section, why the Keffs in FoldME is different from AcidifyME/OxidizeME? The authors states that “This causes more protective chaperones to be produced resulting in additional proteome stress that decreases growth rates at high temperature (Figure 2C) or at low pH (Figure 2D).”, however, model with FoldME keffs requires more protein resources based on Fig. 2E and F.

4. In the Phenotype validation using consolidated kinetome section, the authors states that “it confirms the overall robustness of StressME”. Does it suggest that the StressME is insensitive to gene/reaction addition/deletion and parameter change?

5. In Methods, reference should be provided to clarify “three single stress ME models (FoldME, OxidizeME and AcidifyME) were rebuilt from the same starting point (E. coli K-12 MG1655 ME-model iJL1678-ME) under Python version 3.6.3 based on the published protocols”.

6. Is there a maximum limit for total protein allocation used in the model development? If so, how the authors determined this value and is it kept constant or variable in the different stress conditions tested?

7. What is the computation time of running the StressME model?

8. The software is not available according to the github URL provided.

Minor:

1. The structure of three proteins (tdcD, purT and ackA) should be in SI-4 rather than Table S2.

2. In the Heat adaptation cannot induce cross-protection against acid-involved stress section, reference should be provided for the statement “This shift requires pyruvate formate lyase (tdcE), which is inactive at pH7.”

Reviewer #2: In this paper, the authors present an integrative metabolic view of Ecoli under three different stresses: thermal, oxidative, and acid stress. Each one of these stresses has been previously analyzed and the authors construct a ME model with the capacity to reproduce previous results and extend to new predictions. This reconstruction (STRESSME) has been included in an open-source computational platform, called EcolIME, in which it is possible to reproduce all the results that the authors support throughout the paper. The paper makes a relevant contribution to designing an in silico platform for Ecoli for exploring how it responds under external stress with practical implications in industry. In general, the paper is well-written and clear in its explanations. However, I suggest that before the paper is accepter for publication the authors attend to the following points:

1) Throughout the paper, it is not clear at all what are the differences between the model from STRESSME and the previous one coming from FoldME, AcidifyME, and OxidizeME. Despite the authors indicating the number of genes and reactions for every reconstruction, still lacks how many reactions are included in the last database that shares and excludes reactions. A global landscape of the differences and similarities can be useful for the reader. Based on the text, seems like the model iZY1689-StressME has close similarity in reconstruction concerning the previous version. Why did iZY1689-StressME have a lower number of proteins (1,568 proteins) than the previous one ( for instance 1,578 proteins), some of them were deleted?

2) It should be convenient to evaluate the quality of the reconstruction with tools like MEMOTE (https://www.nature.com/articles/s41587-020-0446-y). This can be useful to identify the quality of the global reconstruction.

3) In terms of the metabolic responses, is it possible the existence different strategies to face the set of stresses? How robust are these mechanisms?

4) How do you deal with the feedback mechanisms between stress and metabolism? To my understanding, this is not included in the model. If included the model can improve their predictions?

4) In terms of equation 1 and the optimization of the Keff vectors. How many parameters are included in the model? Keff represent a vector, one for each reaction in the reconstruction, or only was subjected to a few of them? Will be useful to explain that point. In addition, equation 1 seems a little counterintuitive when including the keff. [Disp-formula pcbi.1011865.e003] indicates that if a Keff value is increased, the amount of the protein to be synthesized is decreased. is it explained by a low effective production of the enzyme that functionally does not perfectly complement its enzymatic work? This assumption is valid for all the enzymes in the reconstruction? please explain.

5) How does keff from the previous model change to the one obtained from STRESSME? The profiles of new keff are unique to obtain the optimization?

6) When the authors say: " we identified incompatibilities between the Keff values used in FoldME and AcidifyME:

swapping their Keff values produced simulations inconsistent with published results for thermal

and acid responses (Figure 2C-D). " How is that possible if the networks seem to be very similar as the authors indicate in the phrase "The resulting model, called iZY1689-StressME, comprises 1,689 genes, 1,578 proteins, 1,673 metabolites,

1,692 complexes and 36,735 reactions (Fig. 1). This biological scope is a significant expansion

over the original iJL1678b-ME composed of 1,678 genes, 1,568 proteins, 1,671 metabolites,

1,526 complexes and 12,655 reactions." is this considered a significative expansion of the previous model? Pls explain.

7) Figure 5, how can be explained the increase of variability as the temperature increases and the systems look for sub-optimal conditions in growth rate? Is there an explanation for this behavior?

8) As the authors argued, StressME is capable of exploring the crosstalk among these external perturbations (thermal, oxidative, and acid stress), however at the end of the paper, the authors explore only two conditions associated with situations where both stimuli are applied by consecutive order. Thinking more about the crosstalk mechanism, what occurs if two or three stresses are applied simultaneously? What are the predictions of the model? Sketch this situation can be useful for the reader.

**Have the authors made all data and (if applicable) computational code underlying the findings in their manuscript fully available?**

Reviewer #1: **No: **The software is not available according to the github URL provided.

Reviewer #2: Yes

PLOS authors have the option to publish the peer review history of their article (what does this mean?). If published, this will include your full peer review and any attached files.

Reviewer #1: **Yes: **Chao Wu

Reviewer #2: **Yes: **Osbaldo Resendis Antonio

Figure Files:

Data Requirements:

Reproducibility:

References:

---

## [Decision Letter · Decision Letter 1]

28 Jan 2024

Dear Dr. Yang,

We are pleased to inform you that your manuscript 'StressME: unified computing framework of Escherichia coli metabolism, gene expression, and stress responses' has been provisionally accepted for publication in PLOS Computational Biology.

Best regards,

Arturo Medrano-Soto, Ph.D.

Guest Editor

PLOS Computational Biology

Mark Alber

Section Editor

PLOS Computational Biology

The reviewers have determined that your mansucript meets the necessary criteria for publication in PLOS Computational Biology.

Reviewer's Responses to Questions

**Comments to the Authors:**

Reviewer #1: My questions have been addressed. I don't have additional ones.

**Have the authors made all data and (if applicable) computational code underlying the findings in their manuscript fully available?**

Reviewer #1: Yes

PLOS authors have the option to publish the peer review history of their article (what does this mean?). If published, this will include your full peer review and any attached files.

Reviewer #1: **Yes: **Chao Wu

---

## [Editor Report · Acceptance letter]

8 Feb 2024

PCOMPBIOL-D-23-01143R1 

StressME: unified computing framework of Escherichia coli metabolism, gene expression, and stress responses

Dear Dr Yang,

I am pleased to inform you that your manuscript has been formally accepted for publication in PLOS Computational Biology. Your manuscript is now with our production department and you will be notified of the publication date in due course.

With kind regards,

Anita Estes
